# Diversity Patterns of Macrofungi in Xerothermic Grasslands from the Nida Basin (Małopolska Upland, Southern Poland): A Case Study

**DOI:** 10.3390/biology11040531

**Published:** 2022-03-30

**Authors:** Janusz Łuszczyński, Edyta Adamska, Anna Wojciechowska, Joanna Czerwik-Marcinkowska

**Affiliations:** 1Institute of Biology, Jan Kochanowski University, 25-420 Kielce, Poland; janusz.luszczynski@ujk.edu.pl; 2Department of Geobotany and Landscape Planning, Faculty of Biology and Veterinary Sciences, Nicolaus Copernicus University, 87-100 Toruń, Poland; adamska@umk.pl (E.A.); ankawoj@umk.pl (A.W.)

**Keywords:** dry grasslands, macrofungi, habitat factors, Ellenberg indicator values

## Abstract

**Simple Summary:**

Southern Poland exhibits a diverse array of habitats for fungi; however, little is known about the richness and diversity of macrofungi occurring in xerothermic grasslands known as dry grasslands. Xerothermic grasslands with their unique flora and fauna are among the most valuable and, at the same time, most severely threatened habitats of Europe’s natural environment. Studies on such habitats were conducted in southern Poland during the period of 2010 to 2013.

**Abstract:**

Macrofungal communities were investigated in seven plant associations of xerothermic grasslands in the Nida Basin located in the Małopolska Upland of southern Poland. Designation of associations at selected study sites was based on phytosociological relevés using the Braun-Blanquet method. During the years 2010–2013, we studied the diversity and distribution of macrofungi in dry grasslands, where 164 species of basidio- and ascomycetes were recovered. We determined the properties of the studied fungal communities and habitat preferences of individual species found in the analyzed xerothermic plant associations using ecological indicators for macrofungi according to Ellenberg indicator values. Diversity patterns of fungal communities in xerothermic grasslands are strongly influenced by various environmental factors. In our study, we focused on recording the fruiting bodies of all macrofungi and the proportion of each species in the study communities, as well as possible identification of the most likely indicator species for particular habitats. We found significant differences for two of the seven associations analyzed, namely *Thalictro**-Salvietum pratensis* and *Inuletum ensifoliae*. However, based on Ellenberg indicator values for fungi, it is not possible to clearly define fungi as indicator species.

## 1. Introduction

Fungi are an essential functional component of terrestrial ecosystems as decomposers, symbionts, and pathogens [1], and they represent one of the most diverse groups of organisms on earth [2]. However, our knowledge of their diversity and ecological function in xerothermic habitats such as dry grasslands is limited. The plant communities of the class *Festuco-Brometea* have a high conservation value and are included in the European network Natura 2000 [3]. Xerothermic grasslands are characterized by a remarkable diversity of rare and protected plant, animal, and fungi, including lichens [4,5,6]. This is due to the habitat conditions—the availability of light and the features of the soils [7]. The information on the biodiversity of macrofungi from such ecosystems is incomplete and scattered across the literature. Fungi are the most important organisms for the degradation of organic and inorganic (e.g., polylactide) matter [8] and play key roles in nutrient and carbon cycling in terrestrial ecosystems as mutualists, pathogens, and free living saprotrophs [9]. Mycorrhizal fungi form symbiotic associations with higher plants, facilitating plant uptake of water and nutrients such as phosphorus and nitrogen, in exchange for photosynthetically fixed carbon [10]. Xerothermic ecosystems are dependent on the activities of saprophytic and mycorrhizal fungi that break down and transport nutrients in the soil. One of the factors that contributes to the success of xerothermic plant communities is their relationship with specialized fungal species [11,12,13]. Fungi can also be used as bioindicators to assess the quality of xerothermic grasslands [13,14]. In addition to their ecological roles, fungi have been used by humans for thousands of years in different ways [15] and are sold in markets worldwide, providing an important source of rural income. Fungi also provide food and habitats for other organisms, and there interactions with other organisms should not be overlooked [16]. For all these reasons, fungi are considered a strategic component in the conservation and management of xerothermic systems [17].

Mróz and Bąba [18] described xerothermic grasslands as stenothermic habitats of a steppe nature, whose occurrence depends on climatic, soil, and orographic conditions. These habitats are primarily found in southern and south-eastern Europe. In Poland, they are most frequently located on sunny slopes with dry and alkaline substrate. Convenient sites may be hillsides, ravines, the steep slopes of river valleys, and permanent talus piles at the base of chalk cliffs. The plant characteristics for this type of phytocoenosis are photophilic and calciphilic. Such plants have adapted to living in dry alkaline or neutral soils, rich in calcium carbonate but poor in organic matter and moisture. Due to the many factors affecting their survival, xerothermic grasslands are among the most threatened plant communities in Europe [18,19,20].

This study seeks to provide information about macrofungal communities regarding their diversity and distribution in southern Poland, through research conducted during the years 2010–2013. Another goal of our study was to correlate the diversity and richness of macrofungi with plant communities and to determine the environmental conditions for the studied plant associations on the basis of fungal occurrence using Ellenberg indicator values [21].

## 2. Materials and Methods

### 2.1. Study Area

The study of macrofungi was conducted in dry (xerothermic) grasslands located in the area of the Nida Basin in southern Poland (Figure 1). The Nida Basin is a vast depression in the Małopolska Upland located between the Świętokrzyskie Mountains and the Kraków-Częstochowska Upland, situated at an altitude of 200–300 m a.s.l. It is a Jurassic syncline filled with upper Cretaceous marls on which a warm, shallow sea appeared in the Tortonian age (Miocene) leaving a series of evaporation deposits [22]. The most typical soil types of the Nida Basin include humus-rich, shallow rendzina soils, formed from carbonate material and exhibiting high soil fertility and a rich assortment of chemical compounds.

The climate of the Nida Basin differs from the surrounding eastern and western parts of the Małopolska Upland. The annual precipitation is lower than in the surrounding regions and amounts to about 540–700 mm [22]. The majority of precipitation (65–69%) normally takes place from April to September, which reflects the continental nature of the basin. The area is characterized by a higher mean annual temperature than the Polish average, ranging between 7.2 and 7.6 °C. The Nida Basin area is one of the warmest areas in Poland. In the southern part of the basin, the mean annual temperature is +7.5 °C. Over the course of the year, as many as 240 warm days were noted, of which nearly 40 had a maximum temperature higher than 25 °C, as well as roughly 6 days with a maximum temperature higher than 30 °C. These high temperatures typically occur from May to September but may also occur in April or October. The annual total precipitation in the area of our study ranges from 550 to 650 mm. Rainfall occurs mainly in the warmer months (April–September). Snowfall occurs on average from 50 to 60 days a year. Snow cover is maintained for an average of 60 days a year, typically remaining from November to March [23].

The xerothermic conditions, prevailing especially on the slopes of gypsum hills with southern and south-western exposure, contribute to the strong heating of soils. The specific microclimate of the Nida Basin results in a vegetation cover that is both unusual for the landscape of Poland and characterized by high biodiversity [22]. Soils on the gypsum hills are covered with xerothermic grass communities (Figure 2), mainly of the *Festuco-Brometea* class, including associations *Sisymbrio-Stipetum capillatae*, *Potentillo-Stipetum capillatae*, *Koelerio-Festucetum rupicolae*, *Seslerio-Scorzoneretum purpureae*, *Thalictro-Salvietum pratensis*, and *Adonido-Brachypodietum pinnati* [22]. Associations were determined by phytosociological relevés using the Braun-Blanquet method. Moss communities also occur, mainly species from the family of *Potinaceae*, together with xerothermic grass interwoven within patches of thermophilic hazel scrubs and small oak forests.

### 2.2. Sampling and Identification of Macrofungi

Field work on fungi in xerothermic grasslands began in October 2010 and was completed in November 2013. Thirty permanent study plots were determined and marked (Appendix A). Their size ranged from 80 to 100 m^2^. The study covered the plant associations: *Adonido-Brachypodietum pinnati*, *Inuletum ensifoliae*, *Seslerio-Scorzoneretum purpureae*, *Thalictro-Salvietum pratensis* (belonging to the *Cirsio-Brachypodion pinnati* alliance), *Sisymbrio-Stipetum capillatae* (belonging to the *Festuco-Stipion* alliance), *Festucetum pallentis* (from the *Seslerio-Festucion duriusculae* alliance), and *Koelerio-**Festucetum rupicolae* (from the *Festuco-Stipion* alliance). Observations and collection of fungi was conducted every two weeks, from March to November of each year. In total, each plot was monitored 15 times per year.

A classification was made of the fungi in terms of their thermal and light requirements as inferred from plant associations in which they occurred [24]. Species from the studied area that were selected for classification primarily included those that are described in the literature as steppe, xerothermic, and thermophilic species. Boundaries were established between thermophilic (two classes) and xerothermic (three classes) species. For this purpose, the relative insolation factor (RI) was used [24]. An RI value of 350 was assigned to species that were distributed across all insolation classes and had uniform percentage of spatial distribution in these classes, meaning they are theoretically neutral regarding insolation [25]. RI values above this threshold denote thermophilic or xerothermic species, while the maximum RI value was 600 for extremely xerothermic species. In the area studied, habitats in insolation classes 4, 5, and 6 dominated, occupying nearly 75% of all the studied area (Table 1).

During every visit, all fruiting bodies of macrofungi were collected and counted, using the tAC parameter, which determines the total number of fruiting bodies of individual species that were recorded on every study plot during the entire study period [26]. While conducting collections, information on the date of collection, type of substrate, number of the plot, and number of fruiting bodies were recorded. Organoleptic features of fungi were also recorded, such as color of cap, stem, and flesh and presence of slime and hydrophanous appearance, as well as odor and taste. In the case of gasteroid fungi, features reflecting shape, size, and color of the endoperidium, color and manner of scaling of the exoperidium, type of peristome, and coloration and construction of the surface of the stem were noted. On each occasion, collected fruiting bodies underwent preliminary cleaning and drying in a dryer at a temperature of 39 °C after being brought back from the field. Laboratory analysis consisted of observation of spores and elements of the hymenium using a light microscope and (in some ascomycete species) a scanning electron microscope (SEM). The light microscope was used to measure length and width of the spores, for observing the sculpture of the epispore (outer coat of ascospores) length and width of the basidia, and length and width of the cystidia. In the case of gasteroid fungi, apart from spores, observations were made on the morphology of the capillitia, which were also measured for diameter of hypha and thickness of surrounding walls. The observation of all structures was conducted using a 40× objective and mounted in anisole (C_7_H_8_O). Scanning electron microscope observations were conducted at Jan Kochanowski University in Kielce and the Scanning Microscope Laboratory at the Department of Biological and Geological Sciences at Jagiellonian University in Kraków. The PCR reaction method was applied for obtaining ITS rDNA sequences (using the protocols of Larsson and Orstadius [27], in order to verify the taxonomic identity of species of genus *Tulostoma.* Moreover, species of the genus *Conocybe* were verified by Anton Hausknecht, University of Vienna (in 2016), and species of the genus *Disciseda* were verified by Gabriel Moreno, University of Alcalá (in 2018).

The permanent observation plots method was supplemented with an on-the-march method between selected permanent plots. At all of the permanent plots, phytosociological relevés were taken using the Braun-Blanquet method [28]. Plant identifications made in the field were verified using the taxonomic keys of Mirek et al. [29] and Matuszkiewicz [30]. Fungal names were used in accordance with Index Fungorum (accessed 1 February 2022) [31] and Wijayawardene et al. [32], and the classification follows Wijayawardene et al. [32] and Hawksworth [33]. Ellenberg indicator values for the fungi followed Simmel et al. [21]. Morphological identification of fungi was made using keys and descriptions provided by Rudnicka-Jezierska [34], Wright [35] and Sarasini [36].

### 2.3. Statistical Analyses

The Shannon Diversity Index H′ was calculated based on the data on the number of fruiting bodies of macrofungi collected in the years 2010–2013 in selected plant communities. These calculations were made in PAST v. 4.02 (Univeristy of Oslo, Oslo, Norway) [37]. To assess whether the H′ values differed in a statistically significant way between plant associations, one-way ANOVA and Tukey post hoc tests were conducted. These analyses were performed with Statistica v. 9.0 software (TIBIC Software Inc., Palo Alto, CA, USA) [38]. Ellenberg indicator values as presented by Simmel et al. [21] for the list of fungi occurring in a given sample were calculated as the mean weighted value of occurrence at a given location. In this way, the following indicators were established: L—light intensity, T—mean annual temperature, F—substrate moisture content, N—substrate nutrient availability, and O—substrate openness according to [21]. Based on these data, indirect ordination analysis PCA (principal component analysis) was conducted.

Data on the quantity and occurrence of fruiting bodies of fungi collected over four years of the study period were used for direct ordination analysis (CCA, canonical correspondence analysis). A file with “environmental data” was constructed based on the year and the plant association in which the study was conducted. This information was encoded in a “0–1” system (binary encoding, also known as dummy variables). In order to determine which of the resulting variables were statistically significant for the variation in fungi occurrence, the forward selection and Monte Carlo permutation tests were performed (Appendix A). The result of the CCA is an ordination diagram in which species are indicated by geometric symbols and environmental variables by vectors. A diagram was also generated which shows changes in diversity index in the ordination plot. PCA and CCA analyses were conducted with Canoco v. 5.0 software (Ithaca, NY, USA) [39]. For the purposes of description and graphic representation of the results, the following abbreviations of plant association names were adopted: AB—*Adonido-Brachypodietum*, Fp—*Festucetum pallentis*, Ie—*Inuletum ensifoliae*, KF—*Koelerio-Festucetum*, SSc—Se*slerio-Scorzoneretum*, SSt—*Sisymbrio-Stipetum*, and TS—*Thalictro-Salvietum pratensis*.

## 3. Results

One hundred and forty-six species of macrofungi collected in seven associations of xerothermic grasslands in the Nida Basin (in the Małopolska Upland of southern Poland) between 2010 and 2013 were identified (Appendix A). The list included 139 basidiomycete species and 7 ascomycete species. These fungi exhibited a clear spatial distribution and relationships with the xerothermic plant associations *Adonido*–*Brachypodietum pinnati*, *Festucetum pallentis*, *Seslerio*–*Scorzoneretum purpureae*, and *Sisymbrio*–*Stipetum capillatae*. In mycological terms, these plant associations can be divided into two groups. The first group comprised associations from the *Festuco*–*Stipion* and *Seslero*–*Festucion duriusculae* alliances which were differentiated by the occurrence of gasteroid fungi. The second group included mesoxerothermic associations from the *Cirsio*–*Brachypodion pinnati* alliance which are characterized primarily by agaricoid fungi. In the first group of the *Festuco*–*Stipion* and *Seslero*–*Festucion duriusculae* alliance, the following species occurred with second and higher degrees of phytosociological stability: *Bovista tomentosa*, *Disciseda candida*, *D. verrucosa*, *Gastrosporium simplex*, *Geastrum campestre*, *G. minimum*, *G. striatum*, *Tulostoma brumale*, *T. pallidum*, *T. kotlabae*, *T. melanocyclum*, and *T. squamosum*. These taxa were recorded exclusively, or nearly exclusively, in the associations *Festucetum pallentis* and *Sisymbrio*–*Stipetum capillatae* which were characterized by the highest frequency of occurrence, number of records, and productivity of fruiting bodies. These fungi formed their own micro-assemblages, not seen in other studied plant associations, and were strongly linked with the most xerothermic plant communities. The main fungal components of these plant communities were species of the genera *Bovista*, *Disciseda*, *Tulostoma*, and *Gastrosporium simplex*, which are all gasteroid species whose construction is best suited to withstand extreme conditions of insolation, air and soil temperature, and soil dryness. These fungi were considered to be local indicator species, characteristic for and differentiating the studied *Seslerio*–*Festucion duriusculae* and *Festuco*–*Stipion* associations.

Among the taxa mentioned, five were also recorded in plant communities of the *Cirsio*–*Brachypodion pinnati* alliance. These were: *Gastrosporium simplex*, *Geastrum striatum*, *Tulostoma brumale*, *T. pallidum*, and *T. melanocyclum*, identified in the *Adonido-Brachypodietum pinnati* association. In the mesoxerothermic *Adonido*–*Brachypodietum* communities, these fungi were recorded only on a single occasion in one of the study plots and thus cannot be considered diagnostically significant for the studied area. The occurrence of *Gastrosporium simplex* in the *Adonido-Brachypodietum* association may be related to its parasitism on the roots of *Stipa capillata* which was also found here. Species which were regularly recorded in the studied rock and *Stipa* communities also included *Mycena pseudopicta* and *Cuphophyllus virgineus*. However, these species were also common in the remaining xerothermic plant associations *Adonido*–*Brachypodietum pinnati*, *Inuletum ensifoliae*, *Seslerio*–*Scorzoneretum purpureae*, and *Thalictro*–*Salvietum pratensis* in the studied area, and thus were characterized by a relatively broad ecological range limiting their value as indicator species. Other taxa of agaricoid fungi, i.e., *Galerina embolus*, *Deconica montana*, *Macrolepiota excoriata*, *Lepista personata*, *Hemimycena mairei*, *Marasmius curreyi*, *Conocybe siennophylla*, *Hygrocybe acutoconica*, and *H. mucronella*, were recorded in the studied area only once during the entire study period; thus, the precise determination of their ecological and habitat requirements, and thus their value as bioindicators, remains uncertain. Based on records of these taxa in the studied plant communities, it can only be stated that they belong either to thermophilic (fungi widespread in terrestrial habitats with minimum temperature of growth at or above 20 °C) or calcareous species.

The Shannon Diversity Index of fungi ranged from 0.41 in the *Thalictro*–*Salvietum* associations to 2.71 in the *Seslerio*–*Scorzoneretum* associations. Differences between H′ values among individual plant communities were statistically significant (Figure 3). Moreover, habitat analysis inferred from indicator values of fungi indicated variations among the studied plots of the same plant association. The most uniform plant communities in terms of the habitat parameters of fungi were *Inuletum ensifoliae* (Ie), *Festucetum pallentis* (Fp), and *Sisymbrio*–*Stipetum* (SSt) (Figure 4). Fruiting bodies within plots of the *Inuletum ensifoliae* were collected from soil with a low level of acidification (based on the indicator values of plants). *Festucetum pallentis* (Fp) was differentiated by higher temperatures and lower soil moisture compared to the other associations. The greatest internal variation was exhibited by plots of *Adonido*–*Brachypodietum* associations. In 2011, a higher level of nutrients associated with greater moisture was noted there, while in 2013 these parameters were considerably lower. Similar dependencies were also noted for *Seslerio*–*Scorzoneretum*, although the fluctuation of these parameters between 2011 and 2013 was lower. The most unfavorable habitat seems to be *Thalictro*–*Salvietum pratensis*, where high moisture levels are accompanied by low temperatures, a low level of nutrients, and a shaded surface.

The remaining fungal species did not show clear preferences in terms of choice of plant community. These species were at the center of the diagram and did not exhibit a correlation with any of the variables which are significant for variation. Nonetheless, this was the quantitatively dominant fungal group, clearly visible in the diagram (Figure 5), where changes in diversity in the ordination plot are shown. Based on the PCA analysis conducted, it can be stated that the habitat conditions of the remaining plant associations are more favorable and more stable, which is providing better possibilities for development resulting in a greater diversity of fruiting fungi. The *Thalictro*–*Salvietum* (TS) associations were significantly less diverse, and poorer in number of species, which may be influenced by less favorable habitat conditions (such as a lower level of nutrients and light).

CCA analysis of direct ordination indicated that significant differences exist between plant communities based solely on differences in the species composition and not on the year in which the studies were conducted. Therefore, to increase the clarity of the ordination diagram (Figure 6), variables associated with the year of study were removed, leaving only those which represented habitat differentiation. The results obtained confirm the dissimilarity of the *Thalictro*–*Salvietum pratensis* and *Inuletum ensifoliae* associations, and the diversity observed is statistically significant. These two variables explain a total of 14.5% of the variability of the dataset. In both of these dissimilar plant communities, species were recorded which did not occur elsewhere, as can be seen in the diagram (Figure 6). In *Inuletum ensifoliae*, these species included *Flammulina ononidis*, *Inocybe dulcamara*, and *Panaeolus foenisecii*, while in *Thalictro-Salvietum*, these species were *Calycina herbarum*, *Hymenoscyphus repandus*, and *Calvatia gigantea*. Additionally, one species, *Clitocybe dealbata*, seemed to prefer the *Inuletum ensifoliae* (Ie) association, but this species occurs also in one other association (*Adonido-Brachypodietum pinnati*), although in smaller number (Appendix A). The situation is similar regarding the species *Lepista luscina*, *Coprinopsis friesii*, and *Entoloma conferendum*, which clearly prefer the *Thalictro-Salvietum* association but were also recorded in the *Adonido*–*Brachypodietum* association.

Following the adopted criteria for thermophilic and xerothermic fungal groups, 12 taxa were included as indicator species (RI value greater than 350, based on at least 10 records). For the remaining species, an RI factor was not calculated. Xerothermic and thermophilic species are grouped according to descending RI factor values, in intervals of 50, resulting in 5 classes: RI 600–551, highly xerothermic; RI 550–501, xerothermic; RI 500–451, moderately xerothermic (Table 2); RI 450–401, thermophilic (Table 3); and 400–351, thermophilic and neutral. To the first, second, and fifth classes, no species were assigned.

These species occur in the warmest habitats with a southern exposure, on dry and shallow soils, on slopes with an incline of 30–40°. They are mainly linked with *Festucetum pallentis* and *Sisymbrio*–*Stipetum* associations.

Thermophilic species are associated with sites located on steep slopes with an incline of 25–30° and a southern exposure, or on even steeper slopes with an exposure close to southern, on dry and shallow soils. They grow accidentally at sites with *Cirsio*–*Brachypodion* vegetation and may not be reliable indicators.

## 4. Discussion

Based on the data obtained in xerothermic grasslands in the years 2010–2013, the distinctiveness of *Inuletum ensifoliae* and *Thalictro*–*Salvietum* plant associations regarding occurrence of macrofungi was demonstrated. *Inuletum ensifoliae* associations were characterized by the occurrence of three species of macrofungi which were not identified in any other habitat, namely *Flammulina ononidis*, *Inocybe dulcamara*, and *Panaeolina foenisecii*, while *Thalictro*–*Salvietum* associations were characterized by *Calycina herbarum*, *Hymenoscyphus repandus*, and *Calvatia gigantea.* One of these species, *Flammulina ononidis*, was identified for the first time in Poland on xerothermic hills near Pińczów town, in associations of *Inuletum ensifoliae* and *Thalictro*–*Salvietum* (Appendix A). This species is supposed to form fruiting bodies in association with roots of *Ononis spinosa* [40]. Habitat preferences of *F. ononidis* at sites in Poland correspond to reports by Bas [41] and Ripková et al. [42], among others. According to these reports, the species was found on limestone sands of dunes and prefers calcareous dunes and river dikes in dry and sunny habitats.

In our study area, numerous species of coprophilous fungi from the genus *Panaeolus* s.l. and *Psilocybe* s.l. were observed. *Panaeolina foenisecii*, belonging to this group, was identified only in the *Inuletum ensifoliae* associations. According to Guzman et al. [43], this species is a cosmopolitan fungus but poorly known with respect to distribution. However, despite the fact that we did not identify *P. foeniseccii* at other sites, it is difficult to confirm its role as an indicator species. At other study sites, we confirmed the occurrence of two additional representatives of the genus *Panaeolus*: *P. olivaceus* and *P. papilionaceus*.

Most hallucinogenic species are found in the genus *Panaeolus* [44,45]. However, the most well-known fungi exhibiting psychedelic effects upon consumption due to their content of psychoactive substances (psilocybin) are members of the genus *Psilocybe* s.l. [46]. In our study, we found seven species that were assigned to the *Psilocybe* complex (Appendix A). Interestingly, in recent years, increasing attention has been paid to the importance of psilocybin and psilocin as therapeutic agents for the treatment of depression, schizophrenia, and autism, for example [47,48,49].

Another noteworthy species of our study was *Schizophyllum commune*, a very common and widespread white mushroom, growing on decaying wood. It has a cap with split gills that roll inward to cover the hymenium in dry weather, which appears to be a xerothermic adaptation. It was recorded in the *Koelerio*–*Festucetum*, *Seslerio*–*Scorzoneretum*, and *Sisymbrio*–*Stipetum* associations, yet not in all years of observation.

As can be seen from analysis of direct ordination, characteristic species for the *Thalictro*–*Salvietum* association are *Calvatia gigantea*, *Calycina herbarum*, and *Hymenoscyphus repandus* (Figure 6). However, it is difficult to establish these fungi as indicator species for the association in which they occur. The fruiting bodies of *Calvatia gigantea* occurred only in the *Thalictro*–*Salvietum* associations during a single season, suggesting that habitat conditions that support fruiting of the species prevailed only one time at a specific site. This species is a terrestrial saprophyte, growing alone or gregariously in meadows, fields, and deciduous forests usually in late summer and autumn [50,51,52]. It is protected in parts of Poland and is of conservation concern in Norway. Furthermore, *Calycina herbarum* and *Hymenoscyphus repandus* are litter decomposing species whose specificity is limited.

Similarly, the occurrence of a species which appears to favor a specific plant community, confirmed by analysis of the direct ordination for the *Inuletum ensifoliae* associations, was *Clitocybe dealbata*. However, the occurrence of this species may also be linked to local climate conditions in the year in which the observation was made.

Finally, preference for specific communities was observed for the species *Coprinopsis friesii*, *Entoloma conferendum*, and *Lepista luscina*, which have been previously reported to occur in the area of the Moszne Lake Reserve by Chmiel [53]. These fungi were found in our study in large numbers in the *Adonido*–*Brachypodietum* association and in fewer numbers in the *Thalictro-Salvietum* association (Appendix A).

The remaining species which we identified during our study did not exhibit a preference in the choice of specific habitats. Examples of such fungi associated with xerothermic grassland include the following fungi: *Gastrosporium simplex* and species from the genus *Tulostoma*, including *T.*
*brumale*, *T. kotlabae*, *T. melanocyclum*, *T. pallidum*, and *T. squamosum* (Appendix A). The majority of these species exhibited an association with xerothermic plant communities, in particular with *Sisymbrio*–*Stipetum capillatae* and *Festucetum pallentis* associations, and were recognized as indicator species for extremely thermophilic and dry grasslands [54]. In addition, the occurrence of these species was also confirmed in xerothermic *Sisymbrio*–*Stipetum* and *Festucetum pallentis* plant associations.

One curious example was the occurrence of a lichenized basidiomycetes (*Lichenomphalia umbelifera*) with agaricoid basidiomata in the analyzed plots, although lichens were not an object of our study. Xerothermic grasslands in Poland belong to rare plant communities with a particular biodiversity of fungi, including lichens. Xerothermic grasslands can be habitat islands and constitute valuable refugia of terrestrial lichens [6]. *Lichenomphalia umbellifera* is common on soil, especially among grasses and mosses (prefers *Sphagnum*) in wet soaks in montane regions. It can be found in the British Isles, Europe, North America, Antarctica, Asia, and Australia [54,55]. The fruiting bodies of *L. umbillifera* showed seasonality, and their formation and durability depended on the climatic conditions in a given year. In drought, fruiting bodies do not develop, and only barren thallus is present [21,56]. This species prefers half-shade (climate value), moderately cold areas, mainly subalpine and upper mountains zones (temperature value), habitats very moist (habitat moisture value), and poor, acidic soils (edaphic value) [56]. We observed the occurrence of *L. umbellifera* (syn. *Omphalina umbellifera* in *Festucetum pallentis*, *Sisymbrio*–*Stipetum*, *Koelerio*–*Festucetum*, and *Inuletum ensifoliae* associations (Appendix A).

The determination of habitat preferences for specific species identified at our study sites allowed to compare and determine the properties of particular plant communities prevailing during the study period. Ecological indicators for macrofungi (similar to those for plants known as Ellenberg indicator values) were compiled by Simmel et al. [21]. Abiotic factors may influence the biology of gastroid fungi of the genera *Tulostoma* and *Geastrum* (among others) associated with xerothermic grasslands [36,53]. For example, *Gastrosporium simplex*, known as the “Steppe Truffle”, or *Tulostoma squamosum* may be indicator species [51]. These species prefer soils with moderate humidity and are strongly nitrophilous, while *Calvatia gigantea*, another gastroid species, is not thermophilic and is neutral to soil pH as well as to temperature. This species grows in sites rich in mineral nutrients, such as abandoned pastures [52]. Similarly, *Clitocybe dealbata* appears to favor a specific association *Inuletum ensifoliae*, where fruiting bodies of this species were identified in one year (Figure 6). However, occurrence may also be linked to local climate conditions in the year in which the observation was made.

Recent molecular studies [57,58] have shown that fungal communities are more diverse than previously known across a range of spatial scales, from local communities to differences across continents. Peay et al. [57] observed that identification of key functional traits is helping to make predictions about the actual diversity of the mycobiome and to decode its role for the health of plants, animals, and ecosystems. On the other hand, Nilsson et al. [58] stated that, “high-throughput sequencing studies of fungal communities are redrawing the map of the fungal kingdom by hinting at its enormous, and largely uncharted, taxonomic and functional diversity.” Therefore, it is important to discuss upcoming trends and techniques, for example, recent results of high-throughput sequencing of fungal communities. There are only a few studies directly comparing fruiting body surveys with molecular data (especially in grasslands). Hay et al. [59] suggest that a combination of both traditional survey and molecular methods gives a good picture. Ovaskainem [60] found that DNA and fruiting body abundance correlated. However, there are also drawbacks to molecular studies of soil samples, such as the fact that they also detect “dead” mycelia, or sampling may not represent the whole site as only a few localized spots can be sampled, which makes it difficult to interpret sequence numbers. Moreover, molecular studies may not be good in predicting the ability to produce fruiting bodies and reproduce. Apart from soil sampling, there also exists spore sampling, which can cover larger areas and time periods. However, even spore sampling may be best applied in conjunction with traditional surveys since the origin of spores cannot easily be traced to the source of origin.

## 5. Conclusions

Fungi play important roles in the ecosystem processes across all terrestrial biomes. Overall, environmental parameters and host plant species can affect fungal community structure in terms of their composition and distribution. Fungi were studied in seven xerothermic grassland associations in southern Poland. A total of 164 species of fungi were recovered. Significant differences were found only for two out of seven analyzed plant associations, namely *Thalictro-salvietum* (TS) and *Inuletum ensifolie* (Ie). In these habitats, fungi not occurring in any other association were found. For Ie, *Flammulina ononidis*, *Inocybe dulcamara*, and *Panaeolina foenisecii*, and for TS, *Calycina herbarum*, *Hymenoscyphus repandus*, and *Calvatia gigantea*. However, an unambiguous determination of the indicator properties of the analyzed fungi still requires further long-term studies. Based on Ellenberg indicator values adopted for fungi, none of the species can be considered a true indicator species at this stage. Although fruiting body surveys do not accurately capture the presence or absence of fungi at given site, they still provide relevant and important information on the ecology of fungi and on plant communities where they are growing.

## Figures and Tables

**Figure 1 biology-11-00531-f001:**
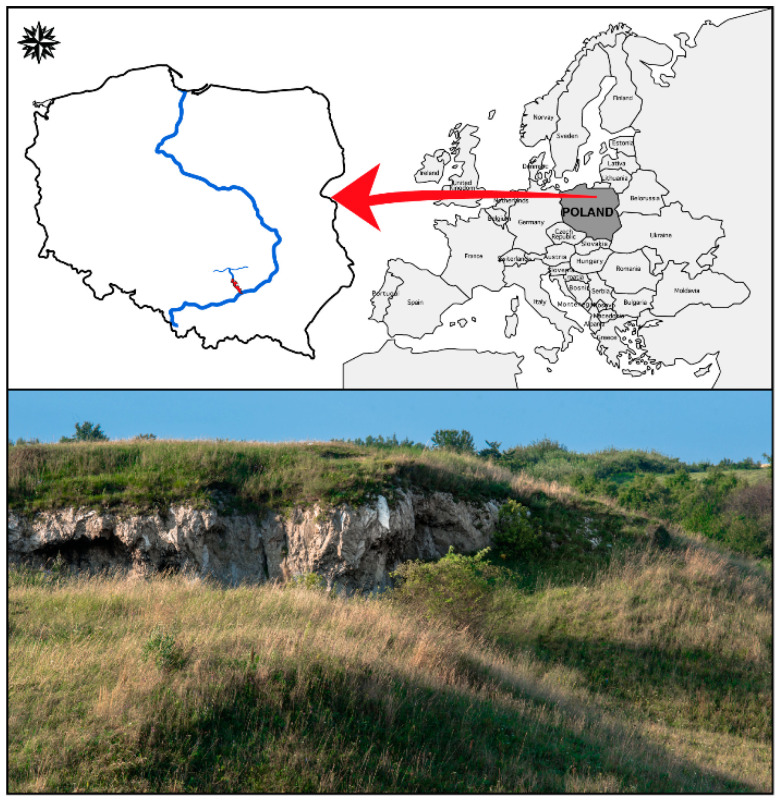
Collection sites in the Nida Basin on map of Poland showing the geographic location of the studied xerothermic grasslands (marked in red).

**Figure 2 biology-11-00531-f002:**
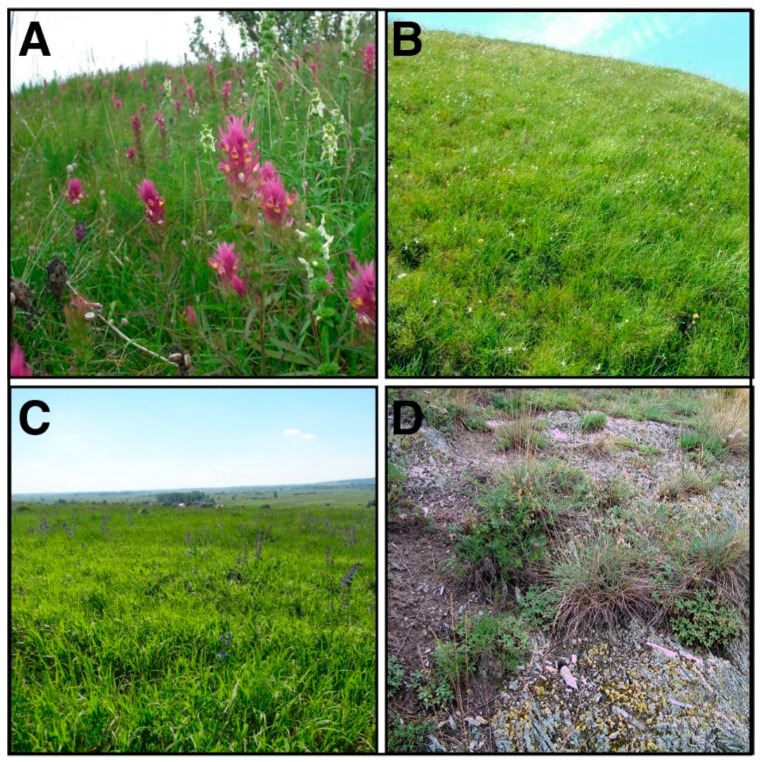
The representative xerothermic grassland habitats in the Nida Basin (southern Poland) where fungi were collected between 2010 and 2013: (**A**) *Adonido-Brachypodietum pinnati*, (**B**) *Seslerio-Scorzonetum*
*purpureae*, (**C**) *Thalictro-Salvietum pratensis*, and (**D**) *Festucetum pallentis*.

**Figure 3 biology-11-00531-f003:**
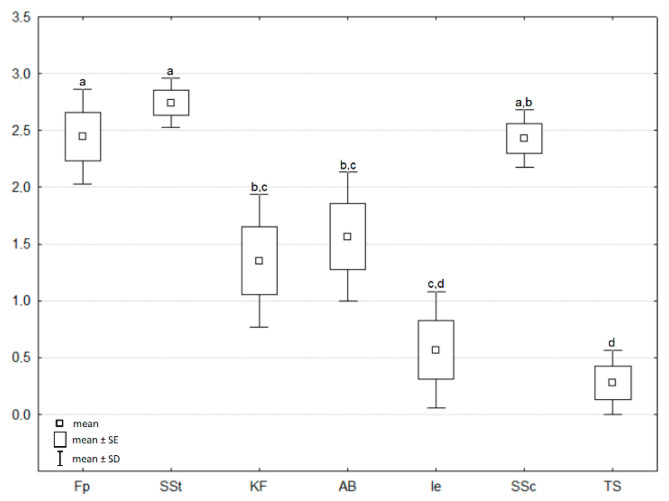
Shannon Diversity Index H′ calculated [±SD and SE] for the dataset of the number of macrofungal species found in the studied plant communities in the years 2010–2013. Letters on the chart indicate the results of one-way ANOVA and post hoc Tukey test analysis. Variables marked with the same letter did not differ from one another significantly, whereas significant differences (*p* < 0.05) occur between groups marked with different letters.

**Figure 4 biology-11-00531-f004:**
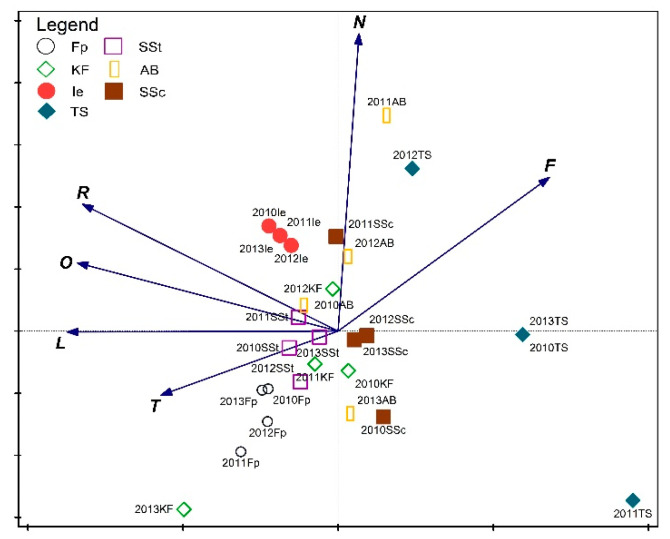
Characteristics of the studied plant associations in the years 2010–2013 with PCA analysis using Ellenberg indicator values of fungi. Legend: AB—*Adonido*–*Brachypodietum pinnati*, Fp—*Festucetum pallentis*, Ie—*Innuletum ensifoliae*, KF—*Koelerio*–*Festucetum*, SSc—*Seslerio*–*Scorzoneretum purpureae*, SSt—*Sissymbrio*–*Stipetum capillatae*, TS—*Thalictro*–*Salvietum pratensis*; indicators: L—light intensity, T—annual temperature, F—substrate moisture content, N—substrate nutrient availability, O—substrate openness, R—substrate reaction (following [21]).

**Figure 5 biology-11-00531-f005:**
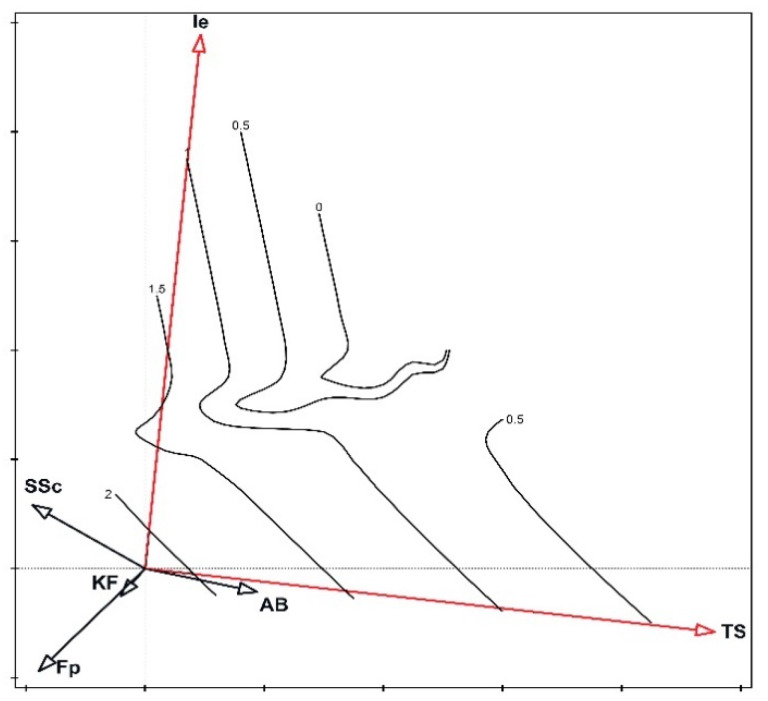
Diversity (H′) in the ordination plot (CCA). Vectors marked in red represent variables that are statistically significant for diversity in the model. Legend: AB—*Adonido*–*Brachypodietum pinnati*, Fp—*Festucetum pallentis*, Ie—*Innuletum ensifoliae*, KF—*Koelerio*–*Festucetum*, SSc—*Seslerio*–*Scorzoneretum purpureae*, TS—*Thalictro*–*Salvietum pratensis*.

**Figure 6 biology-11-00531-f006:**
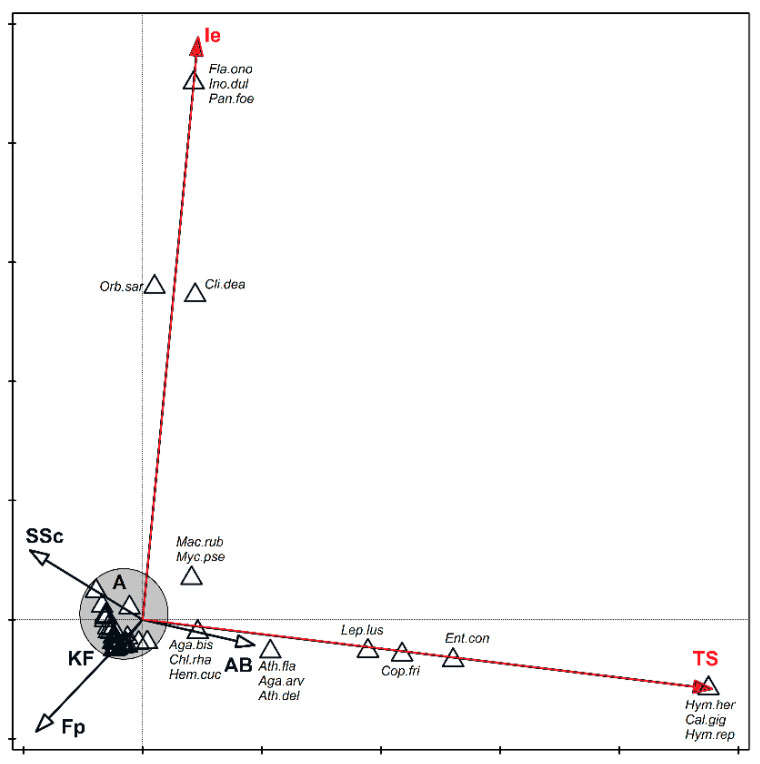
Canonical correspondence analysis (CCA) for the dataset on the occurrence of macrofungal species in studied plant associations. Vectors of variables which are significant for diversity are marked in red. They explain collectively 14.5% of the total variability. Abbreviations of species names are as follows: *Aga.arv*, *Agaricus arvensis*; *Aga.bis*, *Agaricus bisporus*; *Ath.del*, *Atheniella delectabilis*; *Ath.fla*, *Atheniella flavoalba*; *Cal.gig*, *Calvatia gigantea*; *Chl.rha*, *Chlorophyllum rhacodes*; *Cli.dea*, *Clitocybe dealbata*; *Cop.fri*, *Coprinopsis friesii*; *Ent.con*, *Entoloma conferendum*; *Fla.ono*, *Flammulina ononidis*; *Hem.cuc*, *Hemimycena cucullata*; *Hym.rep*, *Hymenoscyphus repandus*; *Lep.cri*, *Lepiota cristata*; *Lep.erm*, *Lepiota erminea*; *Lep.lus*, *Lepiota luscina*; *Mac.rub*, *Macrolepiota rubescens*; *Myc.pse*, *Mycena pseudopicta*; *Orb.sar*, *Orbilia sarraziniana*; *Pan.foe*, *Panaeolina foenisecii*. Abbreviations for associations: AB—*Adonido*–*Brachypodietum pinnati*, Fp—*Festucetum pallentis*, Ie—*Innuletum ensifoliae*, KF—*Koelerio*–*Festucetum*, SSc—*Seslerio*–*Scorzoneretum purpureae*, TS—*Thalictro*–*Salvietum pratensis*. **GROUP A: *Aga.cam***
*Agaricus campestris*, ***Aga.xan** Agaricus xanthodermus*, ***Agr.ped** Agrocybe pediades*, ***Ale.aur** Aleuria aurantia*, ***Arr.gri** Arrhenia griseopallida*, ***Bov.aes** Bovista aestivalis*, ***Bov.lim** Bovista limosa*, ***Bov.nig** Bovista nigrescens*, ***Bov.plu** Bovista plumbea*, ***Bov.tom** Bovista tomentosa*, ***Bov.utr** Bovistella utriformis*, ***Cli.pop** Clitocella popinalis*, ***Cli.agr** Clitocybe agrestis*, ***Con.apa** Conocybe apala*, ***Con.pse** Conocybe pseudocrispa*, ***Con.pub** Conocybe pubescens*, ***Con.sie** Conocybe siennophylla*, ***Con.ten** Conocybe tenera*, ***Cor.gal** Coriolopsis gallica*, ***Cre.epi** Crepidotus epibryus*, ***Cri.sca** Crinipellis scabella*, ***Cru.lae** Crucibulum laeve*, ***Cup.pra** Cuphophyllus pratensis*, ***Cup.rus** Cuphophyllus russocoriaceus*, ***Cup.vir** Cuphophyllus virgineus*, ***Cya.oll** Cyathus olla*, ***Cya.ste** Cyathus stercoreus*, ***Dec.cop** Deconica coprophila*, ***Dec.inq** Deconica inquilina*, ***Dec.mon** Deconica montana*, ***Dis.bov** Disciseda bovista*, ***Dis.can** Disciseda candida*, ***Dis.ver** Disciseda verrucosa*, ***Ent.inc** Entoloma incanum*, ***Ent.sec** Entoloma sericeum*, ***Ent.ser** Entoloma serrulatum*, ***Gal.emb** Galerina embolus*, ***Gal.gra** Galerina graminea*, ***Gal.mar** Galerina marginata*, ***Gal.pum** Galerina pumila*, ***Gal.tri** Galerina triscopa*, ***Gas.sim** Gastrosporium simplex*, ***Gea.ber** Geastrum berkeleyi*, ***Gea.cam** Geastrum campestre*, ***Gea.kot** Geastrum kotlabae*, ***Gea.min** Geastrum minimum*, ***Gea.nan** Geastrum nanum*, ***Gea.pec** Geastrum pectinatum*, ***Geo.are** Geopora arenicola*, ***Gli.lae** Gliophorus laetus*, ***Gym.oci** Gymnopus ocior*, ***Heb.mes** Hebeloma mesophaeum*, ***Hem.cri** Hemimycena crispata*, ***Hem.mai** Hemimycena mairei*, ***Hyg.acu** Hygrocybe acutoconica*, ***Hyg.ing** Hygrocybe ingrata*, ***Hyg.muc** Hygrocybe mucronella*, ***Hyg.par** Hygrocybe parvula*, ***Hym.scu** Hymenoscyphus scutula*, ***Hyp.fas** Hypholoma fasciculare*, ***Ino.ser** Inocybe serotina*, ***Lac.tor** Lactarius torminosus*, ***Len.bru** Lentinus brumalis*, ***Lep.cri** Lepiota cristata*, ***Lep.erm** Lepiota erminea*, ***Lep.ore** Lepiota oreadiformis*, ***Lep.per** Lepista personata*, ***Lep.sor** Lepista sordida*, ***Leu.cre** Leucocoprinus cretaceus*, ***Lic.umb** Lichenomphalia umbellifera*, ***Lyc.der** Lycoperdon dermoxanthum*, ***Lyc.exc** Lycoperdon excipuliforme*, ***Lyc.liv** Lycoperdon lividum*, ***Lyc.mol** Lycoperdon molle*, ***Lyc.per** Lycoperdon perlatum*, ***Lyc.pra** Lycoperdon pratense*, ***Mac.exc** Macrolepiota excoriata*, ***Mac.mas** Macrolepiota mastoidea*, ***Mac.pro** Macrolepiota procera*, ***Mac.ven** Macrolepiota venenata*, ***Mar.cur** Marasmius curreyi*, ***Mar.lim** Marasmius limosus*, ***Mar.ore** Marasmius oreades*, ***Mar.rot.** Marasmius rotula*, ***Mel.mel** Melanoleuca melaleuca*, ***Myc.aet** Mycena aetites*, ***Myc.epi** Mycena epipterygia*, ***Myc.gar** Mycena galericulata*, ***Myc.gal** Mycena galopus*, ***Myc.lep** Mycena leptocephala*, ***Myc.oli** Mycena olivaceomarginata*, ***Omp.pyx** Omphalina pyxidata*, ***Pan.oli** Panaeolus olivaceus*, ***Pan.pap** Panaeolus papilionaceus*, ***Pan.con** Panus conchatus*, ***Pan.rud** Panus rudis*, ***Pez.ves** Peziza vesiculosa*, ***Pha.abi** Phaeoclavulina abietina*, ***Phe.pom** Phellinus pomaceus*, ***Pic.bad** Picipes badius*, ***Pro.sem** Protostropharia semiglobata*, ***Psa.cor** Psathyrella corrugis*, ***Psa.fat** Psathyrella fatua*, ***Psa.pro** Psathyrella prona*, ***Psa.pse** Psathyrella pseudogracilis*, ***Psa.spa** Psathyrella spadiceogrisea*, ***Psi.cor** Psilocybe coronilla*, ***Rho.par** Rhodocybe parilis*, ***Ric.fib** Rickenella fibula*, ***Sch.com** Schizophyllum commune*, ***Scl.gas** Sclerogaster gastrosporioides*, ***Ste.hir** Stereum hirsutum*, ***Str.alb** Stropharia albonitens*, ***Tep.amb** Tephrocybe ambusta*, ***Tra.hir** Trametes hirsuta*, ***Tra.och** Trametes ochracea*, ***Tri.pes** Tricholoma pessundatum*, ***Tri.por** Tricholoma portentosum*, ***Tub.con** Tubaria conspersa*, ***Tub.fur** Tubaria furfuracea*, ***Tul.bru** Tulostoma brumale*, ***Tul.kot** Tulostoma kotlabae*, ***Tul.mel** Tulostoma melanocyclum*, ***Tul.pal** Tulostoma pallidum*, ***Tul.squ** Tulostoma squamosum*, ***Xyl.hyp** Xylaria hypoxylon*. **OTHERS*: Aga.arv*** *Agaricus arvensis*, ***Aga.bis***
*Agaricus bisporus*, ***Ath.del***
*Atheniella delectabilis*, ***Ath.fla***
*Atheniella flavoalba*, ***Cal.gig***
*Calvatia gigantea*, ***Chl.rha***
*Chlorophyllum rhacodes*, ***Cli.dea***
*Clitocybe dealbata*, ***Cop.fri***
*Coprinopsis friesii*, ***Ent.con***
*Entoloma conferendum*, ***Fla.ono***
*Flammulina ononidis*, ***Hem.cuc***
*Hemimycena cucullata*, ***Hym.her***
*Hymenoscyphus herbarum*, ***Hym.rep***
*Hymenoscyphus repandus*, ***Ino dul***
*Inocybe dulcamara*, ***Lep.lus***
*Lepista luscina*, ***Mac.rub***
*Macrolepiota rubescens*, ***Myc.pse***
*Mycena pseudopicta*, ***Orb.sar***
*Orbilia sarraziniana*, ***Pan.foe***
*Panaeolus foenisecii*.

**Table 1 biology-11-00531-t001:** Relative insolation classes in years 2010–2013.

Class of Insolation	% of Insolation	% of Insolated Surface	Values for Stand Quality
1	76–85	6.7	1
2	86–95	3.3	2
3	96–105	16.7	3
4	106–115	23.3	4
5	116–125	26.7	5
6	126–135	23.3	6

**Table 2 biology-11-00531-t002:** Group III. Moderately xerothermic species (with RI values).

Species	RI Values
*Mycena pseudopicta*	500
*Lycoperdon dermoxantha*	483
*Bovista tomentosa*	470
*Geastrum striatum*	470

**Table 3 biology-11-00531-t003:** Group IV. Thermophilic species (with RI values).

Species	RI Values	Species	RI Values
*Cyathus olla*	450	*Gastrosporium simplex*	440
*Geastrum campestre*	450	*Tulostoma squamosum*	440
*Geastrum minimum*	450	*Disciseda candida*	437
*Lycoperdon lividum*	450	*Galerina graminea*	437
*Atheniella flavoalba*	446	*Cuphophyllus virgineus*	433
*Tulostoma brumale*	446	*Disciseda bovista*	433
*Tulostoma melanocyclum*	446	*Marasmius oreades*	420
*Crinipellis scabella*	443		

## Data Availability

The data presented in this study are available in Appendix A.

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
