# Peer review of "Diversity Patterns of Macrofungi in Xerothermic Grasslands from the Nida Basin (Małopolska Upland, Southern Poland): A Case Study"

_biology, 2022, doi:10.3390/biology11040531_

Round 1
Reviewer 1 Report
The revised manuscript is significantly better that the initial version, although it still needs to be gone through by a native English speaker.
Line 19, 187, 260, 277, 401, and any places elsewhere. When the authors say “146 fungi”, I think they mean “146 fungal species”, but many readers will take “146 fungi” to mean “146 fruiting bodies [of an unknown number of species]”. Please consider clarifying this.
- “analyzed” > “recovered”
- “and their diversity and distribution in dry grasslands.” – a verb is missing here. What is intended?
- Please clarify “in xerothermic habitats dry grasslands is limited.”
- “are soil” > “is soil”
- “Fungi can also used” > “Fungi can also be used”
- “They “ > “These habitats” ? Or are the authors referring to Mroz & Baba?
- I propose: “, and permanent talus piles at the base of chalk cliffs. The plant characteristics for”
- “was to” > “is to” since you use “seeks” on 71.
- I propose: “which reflects the continental nature of the basin.”
- “is+7.5” > “is +7.5”
- “results in” > “results in a”
- I propose “relative insolation factor (RI)”
- “was recorded” > “were recorded”
- The concept of “epispore” is introduced very abruptly here. Will all reads know what the authors mean by it? I doubt it, and the authors offer no explanation or definition. Please clarify.
- I propose: “length and width of basidia” … “length and width of cystidia”.
- “capillitium, which were” should be “capillitium, which was” or “capillitia, which were”
- I propose: “in an immersion of”
- I propose: “to verify the taxonomic identity of these specimens.”
- On line 19 the authors write “relevés”. Here they write “relives”. Is the same word intended? If not, what is “relives”?
159 and elsewhere. I’d say there is a double space, “ “, on this line. Please do a Search & Replace.
164, 166, 181. Delete “programme”.
- I propose: “conducted in an attempt to create”
- “during CCA analysis forward selection and Monte Carlo permutation tests we performed during the CCA analysis.”. I’m afraid I cannot follow what was done here. Please clarify this sentence.
- “Tulostoma brumale var. brumale”. This is tautological. Or in what way do the authors mean that “Tulostoma brumale var. brumale” differs from ““Tulostoma brumale” unless Latin names were used in a loose sense? Do the authors, in fact, use Latin names in a loose, informal sense in the manuscript?
- I propose “acutoconica, and “
- “and as follows of” > “and thus”
- Is “calcifying” really the correct word here, “turns to chalk” ?
- “was less” > “was lower”
- “represent” > “represented”
- I propose: “as can be seen in the diagram.”. But what diagram is that? Figure 5?
- “communities” > “community”
- “they were” > “these species were”
- Please clarify “grouped from”
- I propose “the diagram (Figure 6), where”
- I propose “being influenced”
- “which are” > “that are”
287-289. Please clarify this sentence – I’m afraid I cannot follow it. Also, “for whom” > “for which”
- What does “exposure resembling an S” mean? Not S-shaped, I take it?
- I propose: “other habitats, namely”
- I propose: “One of the characteristic species”
- “in the” > “in” ?
- “in its” > “with respect to”
- “contain” > “contains”
- “grows” > “and grows”.
- I propose: “wood. It has a”
- I propose: “Interestingly, this species was described as hallucinogenic in an Australian study”
- “coprophilic” – but Schizophyllum commune grows on trees/wood? See: “It is found in the wild on decaying trees after rainy seasons followed by dry spells where the mushrooms are naturally collected.” (https://en.wikipedia.org/wiki/Schizophyllum_commune)
- I propose: “in that it is common in”
- I propose: “This species is a terrestrial”
- “this communities” > “these communities”
- “a definite communities” > “a specific community”
- I propose: “of the species Coprinopsis”
- I propose: “agaricoid basidiomata” instead of the other way around.
- “In the British isles … Australia [50,51].” This is not a complete sentence – at least one verb is missing.
- “For L. umbellifera evaluated were the” – this makes no sense. At least one verb is missing.
- I propose: “These species prefer soils”
- “definite” > “specific”
- “It is essential is how” – at least one verb is missing.
- “and they were” > “namely”
- “Fungi productivity” could be anything. I think the authors mean “Incidence of macrofungal fruiting bodies”
Author Response
Answers to points raised by the Reviewers of manuscript ID: biology-1603631
"Diversity Patterns of Fungi in Xerothermic Grasslands from the Nida Basin, (Małopolska Upland, Southern Poland): Case Study” by Janusz Łuszczyński et al.
I would like to thank you for the remarks about our manuscript titled “Diversity Patterns of Fungi in Xerothermic Grasslands from the Nida Basin, (Małopolska Upland, Southern Poland): Case Study”. I am grateful to Reviewers for their comments. These comments allowed me to improve the quality of the manuscript. Below, I listed my answers to all questions and points raised by the Reviewers. All changes in revised manuscript are marked in green font.
Answers to the Reviewer’s comments:
|
Comment |
Answer |
|
Reviewer 1 |
|
|
The revised manuscript is significantly better that the initial version, although it still needs to be gone through by a native English speaker. |
Thank you for this comment. Whole manuscript was reviewed by a native English speaker. I hope that it is much better now. |
|
Line 19, 187, 260, 277, 401, and any places elsewhere. When the authors say “146 fungi”, I think they mean “146 fungal species”, but many readers will take “146 fungi” to mean “146 fruiting bodies [of an unknown number of species]”. Please consider clarifying this. |
Thank you for this comment and I apologize for the confusion with the “146 fungi”. I added correction (“species of fungi”). I hope that it is much better now. |
|
Line 20: “analyzed” > “recovered” |
Thank you for this comment. Done. |
|
Line 20: “and their diversity and distribution in dry grasslands.” – a verb is missing here. What is intended? |
Thank you for this comment. I changed this sentence. “During the years 2010-2013 we studied the diversity and distribution of fungi species in dry grasslands, where one hundred and forty-six species of fungi were recovered.” |
|
Line 37: “Please clarify “in xerothermic habitats dry grasslands is limited.”
|
Thank you for this comment. I added new information. “However, our knowledge of their diversity and ecological function in xerothermic habitats dry grasslands is limited (by climatic factors, topographic features, soil conditions and management practices, in particular the intensity of grazing).” |
|
Line 49: “are soil” > “is soil” |
Thank you for this comment. Done. |
|
Line 51: “Fungi can also used” > “Fungi can also be used” |
Thank you for this comment. Done. |
|
Line 58: “They “ > “These habitats” ? Or are the authors referring to Mroz & Baba? |
Thank you for this comment and I apologize for the confusion with “they”. I added “These habitats”. |
|
Line 62: I propose: “and permanent talus piles at the base of chalk cliffs. The plant characteristics for”
|
Thank you for this comment. I changed this sentence. |
|
Line 73: “was to” > “is to” since you use “seeks” on 71.
|
Thank you for this comment. Done. |
|
Line 90: I propose: “which reflects the continental nature of the basin.”
|
Thank you for this comment. I added new phrase according to suggestion. |
|
Line 93: “is+7.5” > “is +7.5”
|
Thank you for this comment. Done. |
|
Line 100: “results in” > “results in a” |
Thank you for this comment. I added “a” in this sentence. |
|
Line 129: I propose: “relative insolation factor (RI)” |
Thank you for this comment. Done. |
|
Line 137: “was recorded” > “were recorded” |
Thank you for this comment. Done. |
|
Line 144: The concept of “epispore” is introduced very abruptly here. Will all reads know what the authors mean by it? I doubt it, and the authors offer no explanation or definition. Please clarify. |
Thank you for this comment. I added new information according to suggestion: Observation using the light microscope was used to measure spores, i.e. in terms of their length and width, and for observation of the sculpture of the epispore (each ascospore is delimited by a pair of unit membranes, the outer of which becomes the spore membrane and the inner the plasmalemma of the spore. The first wall layer, the electrondense epispore, containing acidic mucins, chitin and protein is formed between these membranes), length and width of the basidium, and length and width of the cystidium. |
|
Line 145: I propose: “length and width of basidia” … “length and width of cystidia”. |
Thank you for this comment. Done.
|
|
Line 146: “capillitium, which were” should be “capillitium, which was” or “capillitia, which were” |
Thank you for this comment. Done. |
|
Line 148: I propose: “in an immersion of” |
Thank you for this comment. I added “an”. |
|
Line 151: I propose: “to verify the taxonomic identity of these specimens.” |
Thank you for this comment. Done. |
|
Line 155: “On line 19 the authors write “relevés”. Here they write “relives”. Is the same word intended? If not, what is “relives”?
|
Thank you for this comment and I apologize for the confusion with “relevés”. I changed this word. |
|
Line 159: and elsewhere. I’d say there is a double space, “ “, on this line. Please do a Search & Replace. |
Thank you for this comment. Done. |
|
Line 164, 166, 181: Delete “programme”. |
Done. |
|
Line 171: “I propose: “conducted in an attempt to create” |
Thank you for this comment. Done. |
|
Line 177: “during CCA analysis forward selection and Monte Carlo permutation tests we performed during the CCA analysis.”. I’m afraid I cannot follow what was done here. Please clarify this sentence. |
Thank you for this comment. I changed this sentence. “In order to determine which of the resulting variables were statistically significant for the diversity in fungi occurrence, the forward-selection and Monte Carlo permutation tests we performed during the CCA analysis.” |
|
Line 207: “Tulostoma brumale var. brumale”. This is tautological. Or in what way do the authors mean that “Tulostoma brumale var. brumale” differs from ““Tulostoma brumale” unless Latin names were used in a loose sense? Do the authors, in fact, use Latin names in a loose, informal sense in the manuscript? |
Thank you for this comment and I apologize for the confusion with tautological name. I changed the latin names. |
|
Line 220: I propose: “acutoconica, and.” |
Thank you for this comment. I added a comma. |
|
Line 221: “and as follows of” > “and thus” |
Done. |
|
Line 224: “Is “calcifying” really the correct word here, “turns to chalk” ? |
Thank you for this comment. I changed “calcifying” to “calcareous”. Calcareous grasslands are important hotspots of plant species diversity in central Europe. They contain many rare and endangered plant species and are of strong conservation interest. However, due to land use changes, calcareous grasslands declined significantly in Europe during the last 150 years. |
|
Line 234: “was less” > ”was lower” |
Thank you for this comment. I changed this phrase. |
|
Line 240: “represent” > “represented”. |
Thank you for this comment. Done. |
|
Line 243: I propose: “as can be seen in the diagram.”. But what diagram is that? Figure 5? |
Thank you for this comment. I changed this sentence and I added Figure 5. |
|
Line 246: “communities” > “community” |
Thank you for this comment. I changed it. |
|
Line 251: Please clarify “grouped from” |
Thank you for this comment. I modified this sentence. “These species were at the centre of the diagram, and do not exhibit a correlation with any of the variables which are significant for variation.” |
|
Line 253: I propose “the diagram (Figure 6), where” |
Thank you for this comment. I changed this sentence according to suggestion. |
|
Line 256: I propose “being influenced” |
Thank you for this comment. I added “being”. I hope that it is much better now. |
|
Line 272: “which are” > “that are” |
Thank you for this comment. Done. |
|
Line 287-289: Please clarify this sentence – I’m afraid I cannot follow it. Also, “for whom” > “for which” |
Thank you for this comment. I changed this sentence. “The remaining taxa were less numerous, thus the seven species with a number of five to nine records, the RI values was calculated, but they are only indicative.” |
|
Line 303: What does “exposure resembling an S” mean? Not S-shaped, I take it? |
Thank you for this comment and I apologize for the confusion with “S”. I added “southern” in this sentence. |
|
Line 311: I propose: “other habitats, namely” |
Done. |
|
Line 313: I propose: “One of the characteristic species” |
Done. |
|
Line 317: “in the” > “in” |
Done. |
|
Line 321: “in its” > “with respect to” |
Thank you for this comment. Done. |
|
Line 322: “contain” > “contains” |
Thank you for this comment. Done. |
|
Line 336: “grows” > “and grows” |
Done. |
|
Line 336: I propose: “wood. It has a” |
Thank you for this comment. Done. |
|
Line 337: I propose: “Interestingly, this species was described as a hallucinogenic in an Australian study” |
Thank you for this comment. Done. |
|
Line 340: “coprophilic” – but Schizophyllum commune grows on trees/wood? See: “It is found in the wild on decaying trees after rainy seasons followed by dry spells where the mushrooms are naturally collected.” (https://en.wikipedia.org/wiki/Schizophyllum_commune) |
Thank you for this comment and I apologize for the confusion with “coprophilic”. I deleted this sentence. |
|
Line 347: I propose: “in that it is common in” |
Thank you for this comment. Done. |
|
Line 349: I propose: “This species is a terrestrial” |
Thank you for this comment. I added “a” in this sentence. |
|
Line 351: “this communities” > “these communities” |
Thank you for this comment. Done. |
|
Line 353: “a definite communities” > “a specific community” |
Thank you for this comment. Done. |
|
Line 357: I propose: “of the species Coprinopsis” |
Thank you for this comment. Done. |
|
Line 322: I propose: “agaricoid basidiomata” instead of the other way around. |
Done. |
|
Line 371: “In the British isles … Australia [50,51].” This is not a complete sentence – at least one verb is missing.
|
Thank you for this comment. I added “It can be found in “ in this sentence. |
|
Line 372: “For L. umbellifera evaluated were the” – this makes no sense. At least one verb is missing.” |
Thank you for this comment. I added “It can be found in “ in this sentence. |
|
Line 384: I propose: “These species prefer soils” |
Thank you for this comment. Done. |
|
Line 387: “definite” > “specific” |
Done. |
|
Line 399: “It is essential is how” – at least one verb is missing. |
Thank you for this comment. I modified this sentence. “Overall, environmental parameters and host plant species could affect fungal community structure in terms of their composition and distribution.” |
|
Line 402: “and they were” > “namely” |
Thank you for this comment. Done. |
|
Line 413: “Fungi productivity” could be anything. I think the authors mean “Incidence of macrofungal fruiting bodies” |
Thank you for this comment. I changed this sentence in Table S2. |
|
Reviewer 2 |
|
|
Line 116-121: Distribution of plot numbers across communities? Geographical distribution? |
Thank you for your comments. I added the missing data in Table S1 and changed the number Table S2. There were five plots per grassland type (Table S2 show accumulated counts from five plots). I hope that it is much better now. Table S1: Geographical characteristics of sampling sites for the fungi in xerothermic grasslands from the Nida Basin (Małopolska Upland). |
|
Line 118-121: Even though I’m not really familiar with the nomenclature, I think there is some disagreement about the naming of the levels: According to the original 1962 Braun-Blanquet publication, names ending with -etum are called Assoziation (German). “-ion” is the ending for “Verband”, for which I found the French translation of “Alliance” (https://www.jstor.org/stable/20034509, last page). The current publication usually uses “association” for names ending with “-etum”, but here, “association” is used for names ending with “-ion”, while “community” is used for “-etum”. Am I missing something? |
Thank you for your comments and I apologize for the confusion with “nomenclature”. I added new naming of the levels. Phytosociological identifiers Class: Festuco-Brometea Order: Festucetalia valesiacae Alliance: Festuco-Stipion Sisymbrio-Stipetum capillatae Alliance: Cirsio-Brachypodion pinnati Inuletum ensifoliae, Thalictro-Salvietum pratensis, Adonido-Brachypodietum pinnati, Seslerio-Scorzoneretum purpureae. I used following Mróz, W., Bąba, W. Monitoring of natural habitats. Methodological guide for natural habitat 6210 Xerothermic grasslands (Festuco-Brometea). 2017, Library of environmental monitoring, 3-14. |
|
Line 127: I have difficulties accessing this reference, maybe it might help to explain more in-depth how RI was calculated? |
Thank you for your comment and I apologize for the confusion with old reference (Michalik 1979 [24]). “A classification was made of the fungi in terms of their thermal and light requirements [24]. Species from the studied area were selected for classification, primarily those which are described in the literature as steppe, xerothermic, and thermophilic. Classification of the fungi was conducted on the basis of their spatial distribution and variation in insolation. Boundaries were established between thermophilic and xerothermic species. For this purpose, the relative insolation factor [24] was used (RI). For species which are theoretically neutral regarding variation in relative insolation factor (RI) [25], in other words those which exhibit a uniform percentage of distribution of records in all insolation classes in conjunction with a uniform percentage share of spatial distribution in these classes, an RI value of 350 was established. The maximum RI value was 600, for extremely xerothermic species. In the area studied, habitats in insolation classes 4, 5, and 6 dominated, occupying nearly 75% of all the studied area (Table 1). “ |
|
Line 150/151: This could be more precise. Do I understand correctly that sequences were compared within phylogenetic trees together with published Tulostoma spp. sequences? Was BLAST used? |
Thank you for the comment. The genus Tulostoma is easy to identify to genus level in the field. Precise determination to species level is, however, often difficult, because the characters used are few in number and subtle in nature. Type specimens were studied from herbaria (according to Index Herbariorum and Index Fungorum). Fungi were photographed in situ, dried, and studied in the laboratory under a stereo-microscope. Morphological features are named in accordance with Wright (1987). Studies under SEM were conducted according to the procedure of Moreno et al. (1995). Three ITS and LSU sequences from Hernandez-Caffot et al. (2011) were retrieved from GenBank and included in the data set. DNA extractions were performed using DNeasy Plant Mini kit, PCR reactions, and sequencing were performed as described in Larsson and Orstadius (2008). |
|
Line 155: relevés/releves |
Thank you for the comment. I added the correct name. |
|
Line 161: Maybe: “Statistical analyses”? |
Thank you for the comment. I added “analyses”. |
|
Line 162: How were the records grouped into samples? At the plot level, or at the grassland association level, separated by year? |
Thank you for the comment. The records grouped at the grassland association level, separated by year. |
|
Line 166: Very similar sentence already found on Line 158. |
Thank you for the comment. I changed this sentence. |
|
Line 170: I don’t see exactly how these references are related to the text. Maybe [21]? |
Thank you for your comment and I apologize for the confusion with references. I deleted references [5,36] and I added new []21]. |
|
Line 175: I believe this is also known as binary encoding as "dummy" variables. |
Thank you for your comment. I added new information according to suggestion. “This information was encoded in a “0–1” system (binary encoding also known as dummy variables).” |
|
Line 178: Common sense? Maybe helpful in figure description? |
Thank you for this comment. I changed this sentence. “In order to determine which of the resulting variables were statistically significant for the diversity in fungi occurrence, the forward-selection and Monte Carlo permutation tests we performed during the CCA analysis.” |
|
Line 187: It may be interesting to know the overall number of fruiting body records in the study. |
Thank you for this comment. I changed this sentence. “One hundred and forty-six species of macrofungi, between 2010 and 2013, in seven associations of xerothermic grasslands in the Nida Basin (in the Małopolska Upland of southern Poland) were identified (Table S1).” |
|
Line 225: As far as I can see in Figure 3, the maximum H’ must be around 3, not 0.71. Does this refer to grassland type averages, or the overall range of individual H’ values? |
Thank you for this comment and I apologize for the mistake. I added the correct value of maximum H’ (2.71). This refer to the overall range of individual H’ values. |
|
Line 237: Any statistical tests done? p-values? See above comment on statistics. |
Thank you for this comment. During the ordination analysis, the Monte Carlo permutation test was performed, which indicates the significance of differentiation in the data (for example: for TS p=0.002 and for Ie p=0.044). |
|
Line 241: More details, p-values? See above comment. Also, “diversity” is a bit ambiguous (could be alpha or beta diversity). Similar statements are in the labels of Fig. 5 and 6, especially in the label of Fig. 5 it could be confused with Shannon diversity. |
Thank you for this comment. Figure 5 presents the diversity expressed by the Shannon index located in the ordination space. |
|
Line 262: Seems the wrong way around: “… do not differ from one another in a statistically significant way (p<0.05). Significant differences occur between groups marked with different letters (p>0.05)”. Should be p>=0.05 first, then p<0.05? |
Thank you for this comment and I apologize for the mistake. I changed this sentence.
|
|
Line 353: How about “...appear to favour specific communities”? |
Thank you for this comment. I changed this phrase. |
|
Line 357-359: Seems a bit disconnected from the rest of the discussion, unless I’m missing something. How do the results of Chmiel [49] compare to the results in the current study? |
Thank you for this comment. Chmiel [49] published the first comprehensive work on the ascomycetes found in Poland. “Chmiel [49] observed the occurrence of the species Coprinopsis friesii, Entoloma conferendum and Lepista luscina in the area of the Moszne Lake Reserve, where fruiting bodies of these fungi were found in large numbers in the Thalictro–Salvietum communities but in Adonido–Brachypodietum they occurred in considerably lesser numbers.” |
|
Line 367: “agaricoid basidiomata”? |
Thank you for this comment. Done. |
|
Line 390-L398: I feel that while mentioning the usefulness of molecular studies using high-throughput sequencing is important, this part seems a bit diconnected from the rest of the discussion. |
Thank you for this comment. I added the sentences about molecular studies as suggested by the reviewer. |
|
Line 410: This sentence may sound a bit negative for a last sentence. I tend to think that every monitoring method (including the molecular ones) has its own biases, and traditional fruiting body surveys are still relevant and informative nowadays. |
Thank you for this comment. I added a new sentence. “Fruiting body surveys do not accurately capture the presence and absence of fungi but are still relevant and informative nowadays.” |
|
Edytor |
|
|
English language and style should be carefully reviewed, to which reviewer 1 offered a number of suggestions |
Thank you for this comment. Whole manuscript was reviewed by a native English speaker. I hope that it is much better now. |
|
Another concern was presumed plagiarism. Please make sure that you always use your own wording and not taking whole sentences from published literature. For example, the sentences in the lines 44, 47-49, 54-55 are same to Runiz-Almenara et al. (2019), PeerJ, 7, e8325 |
Thank you for this comment. I changed this sentence. I added new phrase and new reference according to suggestion. Ruiz-Almenara, C.; Gandana, E.; Gomez-Hernandez, M. Macrofungal species between intensive mushroom harvesting and non-harvesting areas in Oaxaca, Mexico. PeerJ 2019, 7:e8325. |
|
Reviewer 3 |
|
|
Line 65: most threatened plant communities in Europe. Why? What are the risk? |
Thank you for this comment. I deleted the earlier sentence and I added a new sentence. “These habitats are primarily found in southern and south-eastern Europe. In Poland, they are most frequently located on sunny slopes with dry and alkaline substrate. Convenient sites may be hillsides, ravines, the steep slopes of river valleys, and permanent talus piles at the base of chalk cliffs. The plants characteristic for this type of phytocoenosis are photophilic and calciphilic. They have adapted to living in dry alkaline or neutral soils, rich in calcium carbonate but poor in organic matter and moisture. Due to the many factors (climatic factors, topographic features, soil conditions and management practices, in particular the intensity of grazing) impacting their survival, xerothermic grasslands are among the most threatened plant communities in Europe. The communities of the class Festuco-Brometea have a high conservation value and are included in the European network Natura 2000 [3]. Xerothermic grasslands are characterized by a remarkable diversity of rare and protected plant, animal, and fungi, including lichens [4–6].” |
|
Line 123: A classification was made of the fungi in terms of their thermal and light requirements. |
Thank you for your comments. I added the missing data in Table S1 and changed the number Table S2. There were five plots per grassland type (Table S2 show accumulated counts from five plots). I hope that it is much better now. Table S1: Geographical characteristics of sampling sites for the fungi in xerothermic grasslands from the Nida Basin (Małopolska Upland). |
|
Line 120: Koelerio-Festucetum belongs to which order of grasslands communities? |
Thank you for your comments and I apologize for the confusion with “nomenclature”. I added new naming of the levels. Phytosociological identifiers Class: Festuco-Brometea Order: Festucetalia valesiacae Alliance: Festuco-Stipion Associations and communities: Association Sisymbrio-Stipetum capillatae Association Koelerio-Festucetum rupicolae Alliance: Cirsio-Brachypodion pinnate Associations and communities: Inuletum ensifoliae, Thalictro-Salvietum pratensis, Adonido-Brachypodietum pinnati, Seslerio-Scorzoneretum purpureae. I used following Mróz, W., Bąba, W. Monitoring of natural habitats. Methodological guide for natural habitat 6210 Xerothermic grasslands (Festuco-Brometea). 2017, Library of environmental monitoring, 3-14. |
|
Line 148: in what cases was it necessary to include SEM pictures for identification? Explain or skip. |
Thank you for this comment. I added new information. “Laboratory analysis consisted of observation of spores and elements of the hymenium using a light microscope and an scanning electron microscope (SEM). Observation using the light microscope was used to measure spores, i.e. in terms of their length and width, and for observation of the sculpture of the epispore (each ascospore is delimited by a pair of unit membranes, the outer of which becomes the spore membrane and the inner the plasmalemma of the spore. The first wall layer, the electrondense epispore, containing acidic mucins, chitin and protein is formed between these membranes), length and width of the basidia, and length and width of the cystidia. In the case of gasteroid fungi, apart from spores, observations were conducted on the morphology of the capillitia, which were also measured for diameter of hypha and thickness of surrounding walls. The observation of all structures was conducted using a dry lens 40x and in an immersion of using anisole (C7H8O). Scanning electron microscope observations were conducted at Jan Kochanowski University in Kielce and the Scanning Microscope Laboratory at the Department of Biological and Geological Sciences at Jagiellonian University in Kraków.” |
|
Line 157: as Index fungorum is a (very) dynamic data base, the date of consultation should be indicated. |
Thank you for your comments and I apologize for the confusion with “Index”. I added new data (accessed 01.02.2022). |
|
Line 210-211: “these fruiting species” |
Thank you for your comments. I changed this sentence. |
|
Line 212: that means, Stipa capillata only occurs in plots, where Gastrosporium simplex was observed? |
Thank you for your comments. I added new information. “The occurrence of Gastrosporium simplex in the Adonido-Brachypodietum was related only to its parasitism on the roots of Stipacapillata. Sisymbrio–Stipetum phytocoenoses contain Stipacapillata, which were also found in Adonido-Brachypodietum communities.” |
|
Line 229: Samples of the Inuletum ensifoliae were collected from soil with low level….. |
Thank you for your comment. I changed this sentence. |
|
Line 232: Is this information based on yearly performed plant sociological relevés? |
Thank you for your comment. Yes, this information based on yearly performed plant sociological relevés. |
|
Line 304: what is a “phytosociological stability”? |
Thank you for your comment. I changed this sentence. |
|
Line 316: “were confirmed” – better: “corresponds to indications” |
Thank you for your comment. I added “corresponds to indications” to phrase. |
|
Line 328-329: psilocybin as therapeutic agent; Skip this information. |
Thank you for your comment. I deleted this sentence. |
|
Line 331-335: Therefor you can skip this paragraph. |
Thank you for your comment. I deleted this paragraph. |
|
Line 336: what was the substrate of your collections of Scgizophyllum commune? |
Thank you for your comment. I added a new sentence. |
|
Line 337: Therefore skip this sentence or mention a possible genetic variability. |
Thank you for your comment. I added a new information and a new reference. |
|
Line 339-340: Do I understand correctly that you count Scgizophyllum commune as a coprophilic species? |
Thank you for your comment. I deleted this phrase and I added a new information. |
|
Line 344: Calvatia gigantea. See: Coetzee &… |
Thank you for your comment. I added a new information and I added a new reference. |
|
Line 404-405: Skip this sentence. |
Thank you for your comment. I deleted this sentence. |
With many thanks for all comments.
Sincerely,
Joanna Czerwik-Marcinkowska

Reviewer 2 Report
Brief summary
The authors studied macrofungal community patterns in the Nida Basin, one of the warmest regions of Poland. Surveys were done on thirty study plots, which they categorized into seven phytosociological types of xerothermic grasslands according to the Braun-Blanquet classification. Every plot was visited 15 times during every of the four study years. Overall, 146 species were identified. Considerable differences in Shannon diversity were observed between the associations. Habitat analysis using part of the species using Ellenberg indicator using PCA indicated that some habitats were more homogeneous than others. Constrained ordination analysis using CCA identified two habitats (Inuletum ensifoliae, Ie uand Thalictro-Salvietum pratensis, TS), which appeared clearly distinct from others. Especially for Ie, a few obvious indicator species were identified. For TS, the picture was a less clear. Furthermore, inferences on the ecological preferences with respect to light/temperature were made for some of the species.
Article
Is the manuscript clear, relevant for the field and presented in a well-structured manner?
I think that the study is valuable and relevant to the field. Given the conservation value of xerothermic grasslands, it is important to learn about associations between species and types of grassland communities, and abiotic conditions in general.
In general, the article is mostly clearly presented and the structure makes sense to me.
Having written this, I'd also like to mention the some difficulties I encountered. Personally not having any background in phytosociology, I found it challenging to keep in mind how the seven associations (names ending with -etum) were related to higher-level groups (-ion). Even though the grouping is mentioned in the Methods section, I wondered whether it may help to show this hierarchical classification in a separate table, along the abbreviations and possibly other information (such as how many study plots were present per association, which seems to be missing from the methods). However, I’m aware that this is very subjective and may be a problem due to my lack of background.
Another (probably rather subjective) comment on the structure of the Results section (authors may disagree): I wonder, if clarity could be improved if starting with a ore global view of the alpha and beta diversity patterns (Shannon H’ and PCA/CCA), and then further going into detail about specific (indicator) species / community associations.
Are the cited references current (mostly within the last 5 years)?
I think there are sufficient current references (even though several are slightly older than 5 years), given that the applied method relies on older literature as well, which is still relevant today in this field.
Does it include an abnormal number of self-citations?
No
Is the manuscript scientifically sound and is the experimental design appropriate to test the hypothesis?
To me, the study design seems appropriate and interesting, especially since the plots were visited every two weeks from March to November, leading to a possibly quite complete survey of the macrofungi at these places. The species identification has been done very carefully using appropriate methods.
It would be interesting to have any information on how the plots were geographically spread (if possible shown on a map). Were the plots located next to each other or far apart? Is it possible that spatial autocorrelation may have played a role?
Another question: how many plots were there per grassland association? Given the overall number of 30, I assume that there were 4-5 per grassland type? Does Table S1 show accumulated counts from these 4-5 plots?
Are the manuscript’s results reproducible based on the details given in the methods section?
Yes, apart from (I think) some missing information about the number of plots per association (see above) and some details about the statistics (see below).
Are the figures/tables/images/schemes appropriate? Do they properly show the data? Are they easy to interpret and understand? Are the data interpreted appropriately and consistently throughout the manuscript?
Yes
Please include details regarding the statistical analysis or data acquired from specific databases.
The overall alpha diversity per grassland association was quantified using the Shannon diversity index, and Tukey’s test was applied post-hoc to find between-group differences. Personally, I would be curious about additionally seeing species accumulation curves of the 15 visits during every year (per plot or per grassland type), and/or optionally presenting some additional estimators of total species diversity apart from Shannon (which is a measure of both species richness and evenness and thus not as straightforward to interpret), e.g. Chao1 or the estimators provided by iNEXT (http://chao.stat.nthu.edu.tw/wordpress/software_download/inext-online). iNEXT also offers an estimate of “sampling completeness”. If sampling achieved a high completeness (due to plots being visited 15 times per year), this could be outlined as a strength of this study.
The PCA ordination of Ellenberg’s indicators and and CCA of the community matrix seem appropriate to me. CCA assumes an unimodal distribution of the response variables (that is, species abundances) depending on the environmental variables (grassland types). Thus, if I understand the theory correctly (not being a statistician!), CCA is appropriate if assuming that species were associated with a single grassland type, but not with several types. Given the focus on mostly indicators for single grassland types, this should be ok. There might be a bias towards sites with more individuals having more weight in the analysis (could be the case given the strong differences in diversity), as well as rare species having a large influence (see https://doi.org/10/d7cp92, p. 279).
From the description in the methods and results section, I did not understand whether the analysis output offered some sort of groupwise comparisons of community composition to underline the distinctiveness of the the fungal Inuletum ensifoliae and Thalictro–Salvietum communities, or whether it was just tested whether there are differences between any of the community types (but no precise information on which ones). Unfortunately I’m not familiar with Canoco, and couldn’t find access to the documentation. I suggest adding the output of the statistical test to the supplement. Relevant statistics such as p-values should be reported in the Results section. It is only mentioned that 14.5% variability was explained. Personally, I would also be interested in how the sampling year variable compared to grassland association in terms of variability explained (in the initial model), even though sampling year was dropped for the final model. If no pairwise comparisons were done, it could be an option to use one of the methods listed here: https://www.researchgate.net/post/Posthoc_test_for_permanova_adonis. However, they are specific for R, I’m not sure about solutions offered by other software packages.
One concern I have regarding the analyses in this paper is the lack of a more elaborate statistical approach at identifying indicator species. The ordination plot from the CCA offers some indication but as far as I understand, no further test for the association of species with grassland types was done. I feel that this would help in further distinguishing more occasional from very strong associations with more confidence, not only having to rely on visual inspection of the patterns. I don’t think that the results of such an analysis would affect the overall conclusions because it seems to me that the authors were careful not overstating the relevance of extremely rare occurrences or occurrences in one single year. They mostly only reported very clear associations as being meaningful. Still, given how important the identification of indicator species is to the paper, I would recommend using some statistical method, and in turn, statements like “appear to favour” could be replaced with something like “were significantly associated with”. As far as I can see, the PAST software used in this study offers the indicator species analysis method from Dufrene & Legendre (1997). Alternatively, the indicspecies R package offers the same (possibly more?) functionality to explore species associations with single communities and even associations with all possible combinations of two or more communities. I assume, that data from every plot and year could be used as a separate replicates, though I'm not a 100% sure if it is a problem having both spatial and temporal replicates in the same analysis.
Are the conclusions consistent with the evidence and arguments presented?
Yes
Please evaluate the ethics statements and data availability statements to ensure they are adequate.
I don’t see any ethical problems (and don’t find any statement on it). Regarding the data, as commented above Table S1 shows accumulated data from several plots, if I’m not mistaken.
Specific comments
L116-121: Distribution of plot numbers across communities? Geographical distribution?
L118-121: Even though I’m not really familiar with the nomenclature, I think there is some disagreement about the naming of the levels: According to the original 1962 Braun-Blanquet publication, names ending with -etum are called Assoziation (German). “-ion” is the ending for “Verband”, for which I found the French translation of “Alliance” (https://www.jstor.org/stable/20034509, last page). The current publication usually uses “association” for names ending with “-etum”, but here, “association” is used for names ending with “-ion”, while “community” is used for “-etum”. Am I missing something?
L127: I have difficulties accessing this reference, maybe it might help to explain more in-depth how RI was calculated?
L150/151: This could be more precise. Do I understand correctly that sequences were compared within phylogenetic trees together with published Tulostoma spp. sequences? Was BLAST used?
L155: relevés/releves
L161: Maybe: “Statistical analyses”?
L162: How were the records grouped into samples? At the plot level, or at the grassland association level, separated by year?
L166: Very similar sentence already found on L158.
L170: I don’t see exactly how these references are related to the text. Maybe [21]?
L175: I believe this is also known as binary encoding as "dummy" variables.
L178: Common sense? Maybe helpful in figure description?
L187: It may be interesting to know the overall number of fruiting body records in the study.
L225: As far as I can see in Figure 3, the maximum H’ must be around 3, not 0.71. Does this refer to grassland type averages, or the overall range of individual H’ values?
L237: Any statistical tests done? p-values? See above comment on statistics.
L241: More details, p-values? See above comment. Also, “diversity” is a bit ambiguous (could be alpha or beta diversity). Similar statements are in the labels of Fig. 5 and 6, especially in the label of Fig. 5 it could be confused with Shannon diversity.
L262: Seems the wrong way around: “… do not differ from one another in a statistically significant way (p<0.05). Significant differences occur between groups marked with different letters (p>0.05)”. Should be p>=0.05 first, then p<0.05?
L353: How about “...appear to favour specific communities”?
L357-359: Seems a bit disconnected from the rest of the discussion, unless I’m missing something. How do the results of Chmiel [49] compare to the results in the current study?
L367: “agaricoid basidiomata”?
L390-L398: I feel that while mentioning the usefulness of molecular studies using high-throughput sequencing is important, this part seems a bit diconnected from the rest of the discussion.
L410: This sentence may sound a bit negative for a last sentence. I tend to think that every monitoring method (including the molecular ones) has its own biases, and traditional fruiting body surveys are still relevant and informative nowadays.
Author Response

(The authors gave the same response as above.)

Reviewer 3 Report
see pdf

Author Response

(The authors gave the same response as above.)

Round 2
Reviewer 2 Report
Thanks for considering my comments on this paper. I’m in general happy with the improvements. I think the English could still be improved in some parts and there are minor mistakes.
I respect the decision to not incorporate any further statistical analyses I suggested (such as rarefaction curves, or indicator species analysis), even though I think the paper would have profited from this. But I suggest that at the minimum, the output of the CCA significance tests should be given in the supplement.
I still have a few comments on specific parts of the paper, which I have added below:
L28: suggestion: “.. significant differences for two of the seven communities (TS and Ie)”. Note: “between” suggests differences only between TS and Ie, but not between TS and other communities or Ie and other communities. I’m still not sure, what would be the best wording… The usual sentences are “variables X and Y were significant in the CCA model”. I also left out the “only”, because it sounds a bit negative, but this may be a subjective choice.
L29: Was it mentioned anywhere in the paper that it was attempted to define indicator species from Ellenberg’s indicator values? Is this an additional finding apart from the descriptive results of the PCA?
L152: “of using” → “of” ?
L183: “Monte Carlo permutation tests we performed” → “were performed”?
L176 (start): improve English
L245 and L249: I have the impression that Figure 6 could be meant at both places instead of Figure 5, or am I wrong? If not, where is Figure 5 mentioned in the text?
L248: In the first review, I suggested that some output of the statistical analysis may be mentioned. Thanks for giving me the p-values in the answers to this comment. However, I would still strongly suggest to include the p-values of the significant associations in the text itself and/or to put the output of the whole analysis (which I assume Canoco offers) in the supplement.
L257: Mix of present and past tense, decide on one.
L269: “do not differ”: I would prefer past tense “did not differ” (but may be subjective). Furthermore, looking at this again I think it is too redundant to mention the significance thresholds of 0.05 twice. Suggestion: remove p>=0.05, then: “Significant differences (p < 0.05) occurred between groups marked with different letters.
L389 end: Ref. 47 instead of 46?
L392: Remove “Rimóczi”
L396: I agree that mentioning molecular studies is good, but I have the feeling that somehow in the current form the text does not integrate well into the "story". Since there isn't any direct connection with the previous sentences, I suggest starting a new paragraph. Furthermore, I wonder whether it may also make sense to be more concrete and directly discuss the pros and cons of fruiting body vs. HTS studies. I'm not very much into the literature here, and there seem to exist only few studies directly comparing fruiting body surveys with molecular data (especially in grasslands), but I found this https://doi.org/10.22621/cfn.v132i4.2027, which also mentiones this: https://doi.org/10.1038/ismej.2013.61. The former suggests that a combination of both traditional survey and molecular methods gives a good picture (which I think could be a good conclusion of the paper). The latter found that DNA and fruiting body abundance correlated. It could be a good idea to finish with a sentence that does not "lower" the value of the current study too much. There are also drawbacks to molecular studies of soil samples, such as the fact that they also detect "dead" mycelia, or it is difficult to sample more than a few localized spots, so a sampling may not represent the whole site, which makes it difficult to interpret sequence quantities (these are anyway subject to PCR amplification and other biases), and molecular studies may not be good in predicting the ability to produce fruiting bodies and reproduce (this last point is a thought of mine, not sure about studies). Apart from soil sampling, there also exists spore sampling (e.g. https://doi.org/10.3389/fevo.2019.00511), which can cover larger areas and time periods. But even spore sampling may be best applied in conjunction with traditional surveys, since the origin of spores cannot easily be known. This is what comes to my mind, the mentioned references may not cover all of it. I hope that there is anything useful in these thoughts.
L408: possibly replace “between” by “for” (see comment on L28)
Author Response
Answers to points raised by the Reviewer 2 of manuscript ID: biology-1603631
"Diversity Patterns of Fungi in Xerothermic Grasslands from the Nida Basin, (Małopolska Upland, Southern Poland): Case Study” by Janusz Łuszczyński et al.
I would like to thank you very much for the remarks about our manuscript titled “Diversity Patterns of Fungi in Xerothermic Grasslands from the Nida Basin, (Małopolska Upland, Southern Poland): Case Study”. I am grateful to the Reviewer for comments. These comments allowed me to improve the quality of the manuscript. Below, I listed my answers to all questions and points raised by the Reviewer. All changes in revised manuscript are marked in blue font.
Answers to the Reviewer’s comments:
|
Comment |
Answer |
|
Reviewer 1 |
|
|
Line 28: suggestion: “.. significant differences for two of the seven communities (TS and Ie)”. Note: “between” suggests differences only between TS and Ie, but not between TS and other communities or Ie and other communities. I’m still not sure, what would be the best wording… The usual sentences are “variables X and Y were significant in the CCA model”. I also left out the “only”, because it sounds a bit negative, but this may be a subjective choice. |
Thank you for this comment and I apologize for the confusion with the “between”. I corrected it and I hope that it is much better now. |
|
Line 29: Was it mentioned anywhere in the paper that it was attempted to define indicator species from Ellenberg’s indicator values? Is this an additional finding apart from the descriptive results of the PCA? |
Thank you for this comment. Yes, it is an additional finding apart from the descriptive results of the PCA. We determined the properties of the studied communities and habitat preferences for the individual species found in the analyzed xerothermic habitats using ecological indicators for fungi according to Ellenberg’s indicators. However, based on Ellenberg’s indicator values, it is not possible to clearly define fungi as indicator species. |
|
Line 152: “of using” → “of” ? |
Thank you for this comment. I deleted “using”. |
|
Line 183: “Monte Carlo permutation tests we performed” → “were performed”? |
Thank you for this comment. Done. |
|
Line 176 (start): improve English |
Thank you for this comment. I changed this sentence. “The comparative visualisation was conducted and the properties of the studied plant communities over the four research seasons was determined.” |
|
Line 245 and Line 249: I have the impression that Figure 6 could be meant at both places instead of Figure 5, or am I wrong? If not, where is Figure 5 mentioned in the text? |
Thank you for this comment and I apologize for the confusion with “Figure 5”. Line 260 present Figure 5. |
|
Line 248: In the first review, I suggested that some output of the statistical analysis may be mentioned. Thanks for giving me the p-values in the answers to this comment. However, I would still strongly suggest to include the p-values of the significant associations in the text itself and/or to put the output of the whole analysis (which I assume Canoco offers) in the supplement. |
Thank you for this comment. I added the CCA significance tests in the supplements Table S2. I modified all numbers of tables. “Table S2: Results of the Monte Carlo permutation tests and forward selection performed during the CCA (canonical correspondence analysis).” |
|
Line 257: Mix of present and past tense, decide on one. |
Thank you for this comment. I changed this sentence. “The remaining species did not show clear preferences in terms of choice of plant community.” |
|
Line 269: “do not differ”: I would prefer past tense “did not differ” (but may be subjective). Furthermore, looking at this again I think it is too redundant to mention the significance thresholds of 0.05 twice. Suggestion: remove p>=0.05, then: “Significant differences (p < 0.05) occurred between groups marked with different letters. |
Thank you for this comment. I modified this sentence according to suggestion. I hope that it is much better now. |
|
Line 389 end: Ref. 47 instead of 46? |
Thank you for this comment. Done. |
|
Line 392: Remove “Rimóczi”
|
Thank you for this comment. Done. |
|
Line 396: I agree that mentioning molecular studies is good, but I have the feeling that somehow in the current form the text does not integrate well into the "story". Since there isn't any direct connection with the previous sentences, I suggest starting a new paragraph. Furthermore, I wonder whether it may also make sense to be more concrete and directly discuss the pros and cons of fruiting body vs. HTS studies. I'm not very much into the literature here, and there seem to exist only few studies directly comparing fruiting body surveys with molecular data (especially in grasslands), but I found this https://doi.org/10.22621/cfn.v132i4.2027, which also mentiones this: https://doi.org/10.1038/ismej.2013.61. The former suggests that a combination of both traditional survey and molecular methods gives a good picture (which I think could be a good conclusion of the paper). The latter found that DNA and fruiting body abundance correlated. It could be a good idea to finish with a sentence that does not "lower" the value of the current study too much. There are also drawbacks to molecular studies of soil samples, such as the fact that they also detect "dead" mycelia, or it is difficult to sample more than a few localized spots, so a sampling may not represent the whole site, which makes it difficult to interpret sequence quantities (these are anyway subject to PCR amplification and other biases), and molecular studies may not be good in predicting the ability to produce fruiting bodies and reproduce (this last point is a thought of mine, not sure about studies). Apart from soil sampling, there also exists spore sampling (e.g. https://doi.org/10.3389/fevo.2019.00511), which can cover larger areas and time periods. But even spore sampling may be best applied in conjunction with traditional surveys, since the origin of spores cannot easily be known. This is what comes to my mind, the mentioned references may not cover all of it. I hope that there is anything useful in these thoughts.
|
Thank you for this comment. I starting “There are only a few studies directly comparing fruiting body surveys with molecular data (especially in grasslands). Hay et al. [55] suggests that a combination of both traditional survey and molecular methods gives a good picture. Ovaskainem [56] found that DNA and fruiting body abundance correlated. However, there are also drawbacks to molecular studies of soil samples, such as the fact they also detect “dead” mycelia, or it is difficult to sample more than a few localized spots, so a sampling may not represent the whole site, which makes it difficult to interpret sequence quantities (these are anyway subject to PCR amplification and other biases), and molecular studies may not be good in predicting the ability to produce fruiting bodies and reproduce. Apart from soil sampling, there also exists spore sampling, which can cover larger areas and time periods. But even spore sampling may be best applied in conjunction with traditional surveys, since the origin of spores cannot easily be known.” [55] Hay, C.R.J.; Thorn, R.G.; Jacobs, C.R. Taxonomic survey of Agaricomycetes (Fungi: Basidiomycota) in Ontario tallgrass prairies determined by fruiting body and soil rDNA sampling. The Canadian Field-Naturalist 2018, 132, 407–424. [56] Ovaskainen, O.; Schigiel, D.; Ali-Kovero, H.; Auvinen, P.; Paulin, L.; Norden, B; Norden, J. Combining high-throughput sequencing with fruit body surveys reveals contrasting life-history strategies in fungi. Microbial Population and Community Ecology 2013, 7, 1696–1709.
|
|
Line 408: possibly replace “between” by “for” (see comment on L28). |
Thank you for this comment. I changed this phrase. |
With many thanks for all comments.
Sincerely,
Joanna Czerwik-Marcinkowska
